# Effects of Extracellular Vesicles Secreted by TGFβ-Stimulated Umbilical Cord Mesenchymal Stem Cells on Skin Fibroblasts by Promoting Fibroblast Migration and ECM Protein Production

**DOI:** 10.3390/biomedicines10081810

**Published:** 2022-07-27

**Authors:** Duc Minh Vu, Van-Tinh Nguyen, Thu Huyen Nguyen, Phuong Thi Xuan Do, Huy Hoang Dao, Do Xuan Hai, Nhi Thi Le, Xuan-Hung Nguyen, Uyen Thi Trang Than

**Affiliations:** 1Vinmec Center for Applied Sciences and Regenerative Medicine, Vinmec Healthcare System, Hanoi 100000, Vietnam; v.ducvm8@vinmec.com (D.M.V.); v.tinhnv8@vinmec.com (V.-T.N.); v.huyennt205@vinmec.com (T.H.N.); dothixuanphuong_t61@hus.edu.vn (P.T.X.D.); huyhoang12082000@gmail.com (H.H.D.); lethinhik62@gmail.com (N.T.L.); v.hungnx1@vinmec.com (X.-H.N.); 2Faculty of Biology, VNU University of Science, Vietnam National University, Hanoi 100000, Vietnam; 3Department of Practical and Experimental Surgery, Vietnam Military Medical University, Hanoi 12108, Vietnam; doxuanhai@vmmu.edu.vn; 4College of Health Sciences, VinUniversity, Hanoi 100000, Vietnam

**Keywords:** umbilical cord-derived, extracellular vesicles, mesenchymal stem cells, skin rejuvenation, fibroblast migration, fibroblast proliferation, extracellular matrix

## Abstract

Umbilical cord-derived mesenchymal stem cells (UCMSCs) have been illustrated for their roles in immunological modulation and tissue regeneration through the secretome. Additionally, culture conditions can trigger the secretion of extracellular vesicles (EVs) into extracellular environments with significant bioactivities. This study aims to investigate the roles of three EV sub-populations released by UCMSCs primed with transforming growth factor β (TGFβ) and their capacity to alter dermal fibroblast functions for skin aging. Results show that three EV sub-populations, including apoptotic bodies (ABs), microvesicles (MVs), and exosomes (EXs), were separated from conditioned media. These three EVs carried growth factors, such as FGF-2, HGF, and VEGF-A, and did not express noticeable effects on fibroblast proliferation and migration. Only EX from TGFβ-stimulated UCMSCs exhibited a better capacity to promote fibroblasts migrating to close scratched wounds than EX from UCMSCs cultured in the normal condition from 24 h to 52 h. Additionally, mRNA levels of ECM genes (COL I, COL III, Elastin, HAS II, and HAS III) were detected with lower levels in fibroblasts treated with EVs from normal UCMSCs or TGFβ-stimulated UCMSCs compared to EV-depleted condition. On the contrary, the protein levels of total collagen and elastin released by fibroblasts were greater in the cell groups treated with EVs compared to EV-depleted conditions; particularly elastin associated with TGFβ-stimulated UCMSCs. These data indicate the potential roles of EVs from UCMSCs in protecting skin from aging by promoting ECM protein production.

## 1. Introduction

Skin aging is gradual progress caused by internal and external factors. Degradative activities happen in both cells and proteins that are components of the extracellular matrix (ECM), which are important for skin morphology and function [1,2]. During the skin aging process, ECM degradation is associated with a decrease in proteoglycans, elastins, and collagens, and an increase in fibril fragmentation, leading to loss of ECM structure integrity [3,4]. Collagen is a principal component of the skin dermis with collagen type I and type III, and is responsible for the tensile strength of the dermis and the reduction of wrinkles [5]. Factors causing cutaneous aging induce collagenase production by human skin fibroblasts, which reduces collagen production and exposes the skin to wrinkles [5]. Another important protein, elastin, is the main protein component of skin tissues that provides elasticity, enhances tensile strength, and plays an important role in tissue repair of wound healing [4]. In addition, hyaluronic acid, which is present in different layers of skin, modulates skin moisture due to its unique capacity to retain water and interact with other components of cutaneous ECM [6]. Therefore, various therapies and products for skin rejuvenation are based on the correction of these ECM components in order to reach a better cutaneous appearance.

Recent studies have demonstrated that exosomes from mesenchymal stem cells (MSCs) promoted skin regeneration and rejuvenation, had anti-aging/wrinkle effects, and inhibited pigmentation in the skin [7,8]. Exosomes (30–250 nm) are a sub-population of spherical, or cup-shaped, extracellular vesicles, besides microvesicles or exosomes (100–1000 nm) and apoptotic bodies (1000–5000 nm), that are enclosed by a lipid bilayer and are secreted by various cell types into the extracellular microenvironment [9]. The therapeutic effect of EVs depends on their cargo, such as miRNAs, proteins, and lipids [9]. Reports have illustrated that growth factors carried by exosomes could alter target cell bioactivities, such as increased proliferation, migration, and protein release, and then induce various cellular behaviors [10,11]. Regarding skin tissue, the beneficial effects of stem cell-derived EVs on reducing skin aging characteristics are that they could promote migration and proliferation of epidermal cells and dermal fibroblasts, and the formation and repair of blood vesicles [12,13]. Kim et al. (2017) showed the effect of cord blood stem cell-derived exosomes (UCBMSC-EXs) on skin rejuvenation by modulating collagen permeability and production [7]. Additionally, exosomes promoted cell migration and increased the production of collagen type I and elastin by fibroblasts and in human skin models [7,14]. These effects indicate that MSC-derived EVs are strong potential candidates for skin rejuvenation and beautification.

Transforming growth factor β (TGFβ), which is a multifunctional cytokine, is a key regulator of ECM assembly and remodeling [15]. Specifically, TGFβ can alter functions, such as proliferation, differentiation, and modulation of immunological properties of MSCs [16,17]. On the other hand, MSCs also secreted TGFβ under the stimulation of various factors, including pro-inflammatory cytokine prime, hypoxia, toll-like receptor 3 agonist, stromal rigidity, and high glucose levels, as reviewed by De Araújo Farias et al. (2018) [17]. The downstream effect of these processes, including a decrease in TGFβ expression by MSCs, is a reduction in ECM protein production, such as α-SMA and COL formation and COL fibers in animal models [18]. Since MSCs are sensitive to 2D culture conditions and mutual relationships between MSCs and TGFβ, we proposed that UCMSCs primed with TGFβ would release extracellular vesicles with influences on ECM protein components of cutaneous tissue. Therefore, in this study, we evaluated the roles of three EV populations, including apoptotic bodies (AB), microvesicles (MV), and exosomes (EX), secreted by UCMSCs under TGFβ stimulation to induce fibroblasts to proliferate, migrate to close wounds and secrete collagen, elastin, and hyaluronic acid. This is important to develop them into potential candidates for skin aging protection.

## 2. Materials and Methods

### 2.1. Umbilical Cord Mesenchymal Stem Cells Isolation and Culture

Primary umbilical cord-derived mesenchymal stem cells (UCMSCs) were supplied by the EV group (Vinmec Center for Applied Sciences and Regenerative Medicine). Cells were expanded to passage five in DMEM/F12 (Gibco, Waltham, MA, USA) supplemented with 10% FBS (Gibco, MA, USA) (*v*/*v*) in T75 cell culture flasks (Nunc, Thermo Scientific, Waltham, MA, USA) surface-coated with CTS^TM^ CELLstart^TM^ substrate (Gibco, MA, USA) under 5% CO_2_ and at 37 °C. Cells were split when they reached 80% confluency using CTS^TM^ TrypLE^TM^ Select Enzyme (Life Technologies, New York, NY, USA).

At P5, UCMSCs were seeded in T75 flask (5000 cell/cm^2^) with EV-depleted culture media (DMEM/F12 supplemented with 10% EV-depleted FBS). EV-depleted FBS was prepared by centrifuging FBS at 120,000× *g*/18 h/4 °C to remove FBS EVs. The UCMSCs were designated as the control when UCMSCs were cultured in normal conditions, and as the cytokine treatment when UCMSCs were treated with 10 ng TGFβ/1 mL culture media for two days. After two days of incubation, UCMSCs from both groups reached 80% confluency, and the conditioned medium was collected for EV isolation.

### 2.2. EV Isolation

Conditioned media collected from cell culture were centrifuged 300× *g*/10 min/4 °C to remove cell debris, then three EV subpopulations, including apoptotic bodies (AB), microvesicles (MV), and exosomes (EX), were collected sequentially at 2000× *g*/20 min/4 °C for AB, 16,500× *g*/30 min/4 °C for MV, and at 100,000× *g*/90 min/4 °C for EX (Optima XPN-100 Ultracentrifuge, Beckman Coulter, Brea, CA, USA). All EVs were resuspended in PBS and stored at −80 °C for further use.

### 2.3. Western Blot

A volume of each EV population was mixed with an equivalent amount of RIPA buffer (Thermo Scientific, CA, USA), incubated for 30 min at 4 °C, and then centrifuged at 16,000× *g* for 20 min at 4 °C. The supernatant was aspirated and transferred to a new 1.5 mL centrifuge tube and stored at −20 °C for further experiments. Protein concentration was determined using the Pierce^TM^ BCA Protein Assay kit (Thermo Scientific, CA, USA) and an optical densitometry method (Optical density-OD) at 560 nm.

Total exosome proteins (15 μg/well) were separated using a 4–12% NuPAGE gel (Invitrogen, Carlsbad, CA, USA) at 200 V for 60 min at 4 °C prior to being transferred to a PVDF membrane (Amersham^TM^, GE Healthcare Life Science, Piscataway, NJ, USA) at 200 mA for 2 h at 4 °C. After that, membranes were incubated with primary antibodies diluted in TBST solution for anti-CD9 (1:100), anti-CD63 (1:200), and anti-GAPDH (1:100) (Abcam, Cambridge, UK) overnight at 4°C. Then, primary antibodies were washed before being incubated with secondary antibody Mouse IgG (Amersham ECL Mouse IgG, HRP-linked whole Ab, GE Healthcare Life Sciences, Piscataway, NJ, USA). Antibody binding was detected by ECL chemiluminescent substrate (Sigma-Aldrich, Singapore) and imaged on an ImageQuant LAS 500 (GE Healthcare Life Science, Piscataway, NJ, USA).

### 2.4. Transmission Electron Microscopy (TEM)

Three EV samples were fixed with 4% paraformaldehyde and then transferred to a carbon grid (Ted Pella Inc., Redding, CA, USA). Samples were then washed before being stained. The EVs were dried at room temperature and photographed by a Transmission Electron Microscopy JEOL 1100 (TEM, JEOL Ltd., Tokyo, Japan) at 80 kV at the National Institute of Hygiene and Epidemiology (NIHE).

### 2.5. Luminex Assay

Growth factors such as epidermal growth factor 2 (EGF-2), hepatocyte growth factor (HGF), and vascular endothelial growth factor A (VEGF-A), were measured by the Luminex assay using ProcartaPlex^TM^ Multiplex Immunoassays (Human Custom ProcartaPlex 3-Plex Kit, ThermoFisher, MA, USA). Frozen EV suspension was thawed and kept on ice for sample preparation, following the manufacturer’s instruction. The luminescent signal was detected using a Luminex^TM^ 100/200^TM^ system with xPONENT 3.1 software (Minneapolis, MN, USA).

### 2.6. Proliferation Assay

Human dermal fibroblasts were seeded into a 96-well plate (2500 cells/well) with EV-depleted media (5% EV-depleted FBS in DMEM/F12) containing 10 μg/mL EVs (AB, MV, or EX) derived from (1) normal UCMSC culture, or (2) TGFβ-stimulated UCMSCs. The EV-depleted medium was used as the negative control (No-EV). Cells were incubated at 37 °C and 5% CO_2_ for 48 h before proliferation analysis using a 3-(4,5-dimethylthiazol-2-yl)-2,5-diphenyl tetrazolium bromide (MTT) assay kit (Abcam, Cambridge, UK). Step by step analysis was performed following the protocol described by the manufacturer. The cell proliferation rate was measured equivalent to the absorbance at 560 nm (SpectraMax M5, Molecular Devices, Silicon Valley, CA, USA) at 0 h and 48 h.

### 2.7. Migration Assay

Human dermal fibroblast cells were seeded at 150,000 cells/well in a 24-well plate to achieve 100% confluency. After achieving cell adhesion, cell proliferation was inhibited by using Mitomycin C (10 μg/mL) for 2 h before making a scratch with a sterile 100 μL tip. Cells were treated with EV-depleted media (5% EV-depleted FBS in DMEM/F12) containing 10 μg/mL EVs (AB, MV or EX) from (1) normal UCMSC cultures, or (2) TGFβ-stimulated UCMSCs. The EV-depleted medium was used as the negative control (No-EV). Cell migration images were observed and captured by inverse microscopy (Canon, Tokyo, Japan) with 10× magnification for different time points. The rates of cell migration to close the wounded areas were analyzed using ImageJ software (Version 1.52P) (National Institute of Heath, Bethesda, MD, USA).

### 2.8. qRT-PCR to Detect Collagen Type I and Hyaluronic Acid

Total RNAs were separated using the TRIzol™ Reagent method (Thermo Fisher Scientific, MA, USA). Purity was then quantified using a Nanodrop^®^ ND-1000 spectrophotometer. cDNA synthesis was performed using the SuperScript™ IV First-Strand Synthesis System Kit (Invitrogen, Vilnius, Lithuania). The cDNA samples were amplified by real-time PCR using the Power SYBR^TM^ Green PCR Master Mix (Thermo Fisher Scientific, MA, USA) on an Applied Biosystems 7500 Real-Time PCR system with parameters of 3 min for amplification, and 40 cycles (95 °C/15 s, 60 °C/45 s). Specific oligonucleotide primers were used to detect collagen type I (COL I), COL III, hyaluronic acid synthase 2 (HAS II), HAS III, elastin, and GDPDH mRNA (Table 1).

### 2.9. Elastin Detection Assay

The secretion of elastin by fibroblasts was determined using a Fastin™ Elastin Kit (Biocolor, Northern Ireland, UK). Cells were seeded in a 12-well plate with 100,000 cells/well density and maintained in different experimental settings similar to the proliferation and migration assay. Cells were incubated at 37 °C in 5% CO_2_ for 48 h before processing according to the manufacturer’s instructions. Briefly, cells were released using CTS^TM^ TrypLE^TM^ Select Enzyme (Thermo Scientific, MA, USA) and maintained in 300 μL of supernatant after centrifugation. Subsequently, an equal volume of Elastin Precipitating Reagent was added to each sample tube. The supernatant was discarded after centrifugation, and the pellets were resuspended in 1 mL Dye Reagent and incubated for 90 min. Pellets were then collected and resuspended in 250 μL of Dye Dissociation Reagent. A volume of 100 μL of suspension was transferred to a 96-well plate for reading OD at 513 nm using spectrometer SpectraMax M3 (Molecular Devices, CA, USA).

### 2.10. Collagen Detection Assay

Total collagen secreted by fibroblasts was quantified using a Sirius Red Total Collagen Detection Kit (Chondrex, Washington, DC, USA). Briefly, cells were seeded in a 6-well plate at a density of 32,000 cells/well and maintained in different experimental conditions similar to the proliferation and migration assay. After reaching 90% confluency, cells were collected and processed as recommended by the manufacturer. A volume of 100 μL of Sirius Red Solution was added to the sample tubes, and pellets were collected after centrifugation. The pellets were rinsed once with washing solution and then resuspended in 250 μL of extraction buffer. The solution was transferred to a 96-well plate for reading OD at 510 nm using a spectrometer (SpectraMax M3; Molecular Devices, CA, USA).

## 3. Results

### 3.1. Characteristics of EVs Originating from TGFβ-Stimulated UCMSCs

In order to stimulate UCMSCs to secrete EVs into the extracellular environment, we incubated UCMSCs with TGFβ (10 ng/mL cell culture media). After two days of cytokine incubation, conditioned media was collected for EV isolation. At the time of conditioned media collection, UCMSCs primed with TGFβ maintained the typical morphology of MSCs with fibroblast-like shapes (Figure 1A).

Additionally, three EV sub-populations were separated, including ABs, MVs, and EXs. These vesicles were investigated for their markers, such as CD9, CD63, and internal control of GAPDH. The expression of GADPH and CD9 occurred in all three EV sub-populations, while CD63 was detected in only the EX fraction (Figure 1B). Regarding EV morphology and size, AB exhibited a rough surface and heterologous morphology with a size of approximately 2 μm, and MVs exhibited a smaller vesicle size from 100 nm to 300 nm (Figure 1C). Finally, EXs had a cup-shaped morphology, and the smallest size from 50–200 nm (Figure 1C).

### 3.2. Growth Factor Expression in EVs Released by TGFβ-Stimulated UCMSCs

To investigate growth factors that affect healthy skin structure, we quantified growth factors of FGF-2, HGF, and VEGF-A using the Luminex assay. Results showed that in the AB population, there was no difference among the three factors analyzed in the same conditions, or the two conditions of normal UCMSCs and TGFβ-stimulated UCMSCs (Figure 2A). In the MV population, HGF was present in a greater amount compared to VEGF-A and FGF-2 in both MVs from normal UCMSCs and TGFβ-stimulated UCMSCs. However, there was no difference in HGF and VEGF-A between MVs derived from normal UCMSCs and MVs derived from TGFβ-stimulated UCMSCs. FGF-2 was higher in the control group of normal UCMSC-derived MVs than in MVs from TGFβ-stimulated UCMSCs (Figure 2B). Similar to the MV population, HGF was expressed in the greatest amount in EXs from both normal UCMSCs and TGFβ-stimulated UCMSCs compared to VEGF-A and FGF-2. VEGF-A was expressed higher in EXs from TGFβ-stimulated UCMSCs compared to the control (Figure 2C). In summary, HGF was expressed the greatest amount in MVs and EXs, and FGF-2 had higher expression in MVs from the control, but VEGF-A had higher expression in EXs from TGFβ-stimulated UCMSCs.

### 3.3. Capacity of EVs from TGFβ-Stimulated UCMSCs in Inducing Fibroblast Proliferation and Migration

In order to investigate the influences of EVs on fibroblast activities such as proliferation and migration, dermal skin fibroblasts were treated with EVs from TGFβ-stimulated UCMSCs and normal UCMSCs (control group–CT group). The EV-depleted fibroblast culture media was used as the negative control (NoEV). Proliferation results showed that EVs, including ABs, MVs, EXs, from TGFβ-stimulated UCMSCs seemed to have a higher capacity to induce fibroblast proliferation, but this was not significant compared to the normal EV (CT group) and negative control (NoEV). Only EXs from normal UCMSC culture (CT group) expressed higher capacity to promote dermal skin fibroblast proliferation than in the NoEV group (*p* < 0.001) (Figure 3A).

Regarding cell migration, data showed that there was no difference in the capacity of AB and MV populations to promote fibroblast migration, except that ABs from the negative control (NoEV) expressed a greater healing rate compared to ABs from TGFβ-stimulated UCMSCs at 24 h (Figure 3A,B). In terms of EX population, the negative control (NoEV-EV-depleted culture media) exhibited the highest capacity to induce dermal fibroblast migration from 24 h to 52 h. In detail, at the 24 h-time point, the NoEV group showed the highest migration rate to close the wounds (*p* < 0.01 and *p* < 0.0001), and EXs from TGFβ-stimulated UCMSCs showed a greater induction of cells migrating to close the wounds compared to the EXs from normal UCMSCs (*p* < 0.05) (Figure 3D). At 44 and 52 h-time points, EXs from TGFβ-stimulated UCMSCs had a similar capacity to the NoEV in inducing fibroblast migration, while EXs from UCMSCs still showed the lowest induction to cell migration (*p* < 0.05 and *p* < 0.001) (Figure 3D). However, no EV groups expressed any different induction to cell migration at the 62 h time point. These data indicate that EVs from normal UCMSCs inhibited dermal fibroblast migration for 52 h.

### 3.4. Capacity of EVs from TGFβ-Stimulated UCMSCs to ECM Gene Expression by Dermal Fibroblasts

As collagens (type I and III), elastin and hyaluronic acid are the main factors responsible for tensile strength, elasticity, and smooth skin, we investigated mRNA expression levels of collagen type I (COL I), COL II, elastin, and hyaluronan synthase type II (HAS II), and HAS III under the induction of EVs from TGFβ-stimulated UCMSCs, normal MSCs (the control), and negative control (NoEV) samples. Results show that the expression of all genes, including COL I, COL III, Elastin, HAS II, and HAS III, was lower in fibroblasts treated with EVs from both normal UCMSCs and TGFβ-stimulated UCMSCs compared to the negative control (NoEV–fibroblasts treated with EV-depleted culture media) (Figure 4). Additionally, there was no difference in mRNA levels of all analyzed genes between fibroblasts treated with EVs from normal UCMSCs or TGFβ-treated UCMSCs; except that the COL III mRNA level was higher in fibroblasts treated with ABs from TGFβ-stimulated UCMSCs than in ABs from normal UCMSCs, and the elastin mRNA level was higher in fibroblasts treated with EXs from normal UCMSCs than EXs from TGFβ-stimulated UCMSCs (Figure 4B,C). These data indicate that EVs from UCMSCs under normal culture conditions or TGFβ stimulation seem to inhibit the mRNA levels of COL I, COL II, elastin, HAS II, and HAS III.

### 3.5. Capacity of EVs from TGFβ-Stimulated UCMSCs to Collagen and Elastin Protein Production by Dermal Fibroblasts

Together with mRNA level analysis of skin structure proteins, we examined the protein production of these genes by fibroblasts. Regarding protein production induced by EVs, we investigated the secretion of elastin and total collagen by dermal skin fibroblasts under three conditions: (1) EVs from TGFβ-treated UCMSCs, (2) EVs from normal UCMSCs, and (3) NoEV (EV-depleted media). Results indicate that there was no difference in the number of proteins detected from three EV sub-populations from TGFβ-stimulated UCMSCs and normal UCMSCs (Figure 5A,B). This means that the three EV sub-populations and EVs, even from TGFβ stimulation or the normal condition, had a similar capacity to stimulate dermal skin fibroblasts to secrete collagen and elastin. However, while MVs from the normal UCMSC culture condition (CT group) induced total collagen production greater than the NoEV group, EXs from TGFβ-stimulated UCMSCs could also induce the collagen production greater compared to the NoEV group. Interestingly, the amount of elastin was greater than in the group of fibroblasts treated with all EVs from TGFβ-stimulated UCMSCs compared to EV-depleted media (Figure 5B). This indicates that all EVs from the cytokine stimulation condition had the capacity to induce dermal fibroblasts to secrete elastin, but only EXs possessed the capacity to induce fibroblasts secreting collagen into the extracellular environment.

## 4. Discussion

It has been reported that priming culturing cells with chemical agents can generate EVs with particular functions [19]. This study used TGFβ to promote UCMSCs in 2D culture to secrete three EV populations, including ABs, MVs, and EXs. These three EV sub-populations exhibited the typical morphology, size, and markers of vesicles as described in previous studies [20,21,22]. As TGFβ plays a role in modulating MSC functions [16,17,23], it is expected that this factor can induce UCMSCs to secrete EVs enveloping different molecules that have functions in dermal fibroblasts. Therefore, we investigated three growth factors of FGF-2, HGF, and VEGF-A in three EV populations released by TGFβ-primed UCMSCs. Unfortunately, we did not find any different expressions of three FGF-2, HGF, and VEGF-A in ABs, MVs, and EXs released from normal MSCs or TGFβ-primed UCMSCs. The only differences were FGF-2 in MVs from normal MSCs being greater than MVs from TGFβ-primed UCMSCs, and VEGF-A in EXs from TGFβ-primed UCMSCs being greater than EXs from normal MSCs (Figure 2). The detection of these factors in EXs released by MSCs from different tissues was reported previously [20], but the different levels of these factors among three EV sub-populations is reported for the first time in the current study. Additionally, these data provide evidence that MVs released by UCMSCs from normal culture conditions may be better for fibroblast function, but EXs released by TGFβ-stimulated UCMSCs are better for inducing tube formation by endothelial cells. This suggestion requires further investigation, both in in vitro and in vivo models.

How these EVs can affect dermal fibroblasts is an important question concerning their use as potential agents for cutaneous wound healing and prevention of skin aging. We examined dermal fibroblast activities within many aspects of proliferation, migration, mRNA and protein expression. EVs from both normal UCMSCs and TGFβ-stimulated UCMSCs did not show a precise proliferation rate of fibroblasts (Figure 3A). This was also observed in ABs and MVs to induce fibroblast migration (Figure 3B,C), but EXs from TGFβ-stimulated UCMSCs expressed a greater capacity to induce cell migration compared to EXs from normal MSCs until 52 h. Previously, three ABs, MVs, and EXs originating from keratinocytes were reported to be associated with fibroblast migration using a bioinformatics assay, but an unclear capacity to induce fibroblast migration was found in in vitro tests [24]. There has not been any report of a direct comparison of three ABs, MVs, and EXs from MSCs, despite different reports on apoptotic bodies, microvesicles, and exosomes in altering target cell functions separately [20,25,26]. The unclear trend in modulating cell proliferation and migration may come from the physiological properties of the donor’s tissues, and the nature of these EVs does not differ from others, except that EXs are distinct from the others. Additionally, the conditioned medium from MSC cultures has been reported as having a therapeutic role in cell proliferation and migration and ECM protein production [8], but this was not seen in the current study. Therefore, there should be more comparisons in roles of EVs and their conditioned media to understand components affecting the bioactivities of the treated cells.

As mentioned, collagen, elastin, and hyaluronic acid are important components required for a young skin appearance. We quantified mRNA levels of these proteins in order to understand which EV populations, such as ABs, MVs, or EXs, are able to promote gene expression. Surprisingly, no EVs promoted the expression of COL I, COL III, Elastin, HAS II, and HAS III in fibroblasts (Figure 4), even though they showed a lower capacity to induce such gene expression compared to the negative control (NoEV treatment). This is contrary to a report by Li et al. (2016) that UCMSCs treated with a low level of TGFβ (0.1 ng/mL) could increase the COL I gene in UCMSCs [27]. This distinction may be due to a different TGFβ dose of 0.1 ng/mL and 10 ng/mL and target cells of UCMSCs and dermal fibroblasts, respectively, as investigated by Li et al. (2016) and in the present study [27]. Thus, data from this study may indicate the inhibition of ABs, MVs, and EXs from all normal MSCs and TGFβ-stimulated UCMSCs in the transcription of these ECM genes. However, trends were observed in protein assays in which collagen and elastin amounts were greater in fibroblasts treated with EVs, especially elastin secreted by fibroblasts treated with TGFβ-stimulated UCMSCs (Figure 5). Collagen protein was produced in larger amounts by dermal fibroblasts under stimulation of exosomes from umbilical cord blood-derived MSCs both in vitro and in a skin model ex vivo [7], which is similar to our results. The inverse expression of mRNA and protein levels of ECM components may reflect a hidden mechanism that promotes ECM protein production, particularly collagen, for example, microRNAs that can participate in the regulation of ECM protein translation. However, this hypothesis of mechanism modulating the expression of ECM mRNAs and protein production from this study requires more careful investigation. Additionally, elastin was released by dermal fibroblasts much greater than with NoEV stimulation, particularly from ABs, MVs, and EVs originating from TGFβ-stimulated UCMSCs. These provide beneficial characteristics of EVs associated with TGFβ stimulation in the correction of skin aging by elastin production.

## 5. Conclusions

In summary, we found that TGFβ-stimulated UCMSCs released three EV sub-populations, including ABs, MVs, and EXs, carrying FGF-2, HGF, and VEGF-A. In general, these cytokine-stimulated EVs did not affect fibroblast proliferation and migration, but cytokine-stimulated EXs showed a greater capacity to induce fibroblast migration. Even though gene expression of cutaneous ECM components was not altered by EVs, total collagen and elastin were increased by the stimulation of EVs from TGFβ-stimulated UCMSCs. Data from this study are important in the further development of EVs released by UCMSCs and TGFβ-stimulated UCMSCs for skin rejuvenation.

## Figures and Tables

**Figure 1 biomedicines-10-01810-f001:**
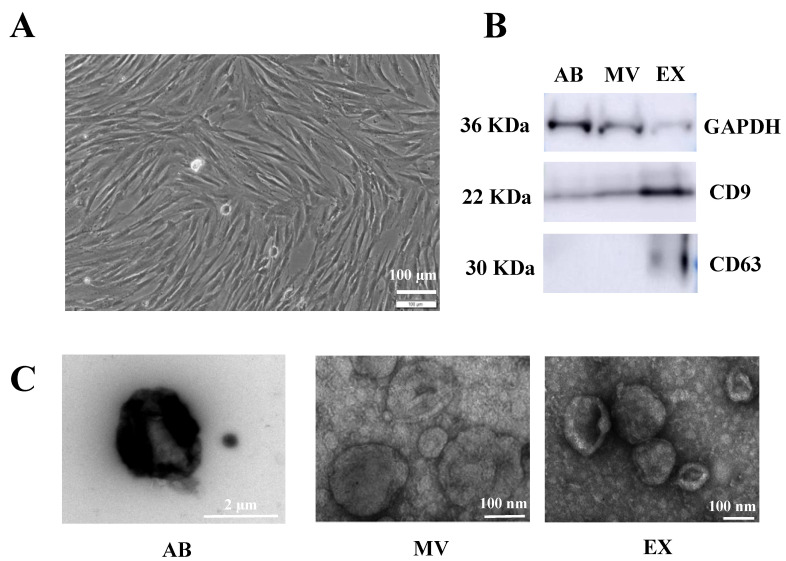
Characteristics of EVs secreted by TGFβ-stimulated UCMSCs. (**A**) Morphology of TGFβ-stimulated UCMSCs. (**B**) Marker expression of EVs: GAPDH and CD9 were detected in all AB, MV, and EX; CD63 was detected in only EX. (**C**) Morphology of EVs under TEM analysis. AB: Apoptotic body, MV: Microvesicle, EX: Exosome.

**Figure 2 biomedicines-10-01810-f002:**
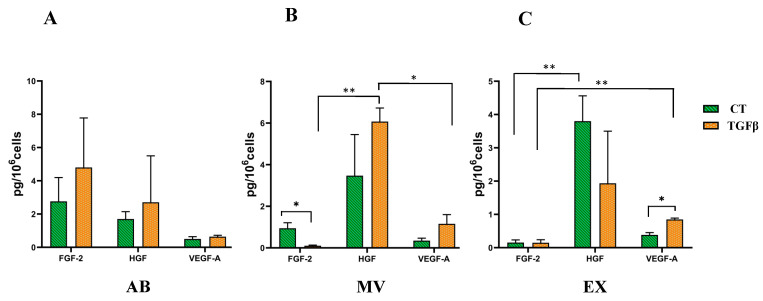
Quantification of growth factors including FGF-2, HGF, and VEGF-A using Luminex assay. (**A**) AB population in which there was no difference in all three factors between the two conditions. (**B**) MV population in which the highest levels of analyzed growth factors belong to HGF, and FGF-2 was more highly expressed in MVs from the control group. (**C**) EX population in which HGF was present in the greatest amount compared to FGF-2 and VEGF-A, and VEGF-A was expressed higher in the TGFβ-stimulated UCMSC group. CT: control group (UCMSCs cultured in normal condition), TGFβ: TGFβ-stimulated UCMSC group, AB: apoptotic bodies, MV: microvesicles, EX: exosomes. N = 3, * indicates *p* < 0.05; ** indicates *p* < 0.01.

**Figure 3 biomedicines-10-01810-f003:**
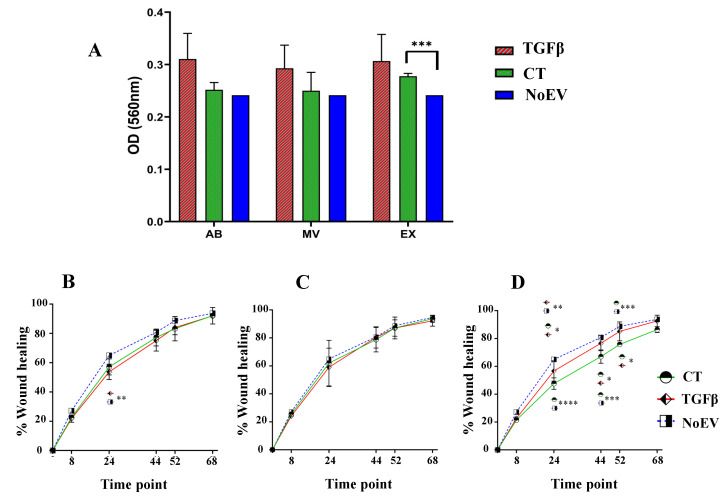
Induction of fibroblast proliferation and migration by EVs from TGFβ-stimulated UCMSCs and normal UCMSCs. (**A**) A higher capacity of EXs derived from normal UCMSCs to induce dermal fibroblast proliferation compared to the negative control (NoEV). (**B**) AB population from the NoEV group stimulated fibroblast migration faster to close wounds compared to ABs from TGFβ-stimulated UCMSCs at 24 h. (**C**) MVs from all three groups did not show any difference in inducing fibroblast migration at all time points. (**D**) Different capacity of EXs to stimulate fibroblast migration: the strongest induction belonged to NoEV group, followed by the EXs from TGFβ-stimulated UCMSC. CT: control group (UCMSCs cultured in normal condition), TGFβ: TGFβ-stimulated UCMSC group, NoEV: EV-depleted cell culture media, AB: apoptotic body, MV: microvesicle, EX: exosome. N = 3, * indicates *p* < 0.05, ** indicates *p* < 0.01, *** indicates *p* < 0.001, **** indicates *p* < 0.0001.

**Figure 4 biomedicines-10-01810-f004:**
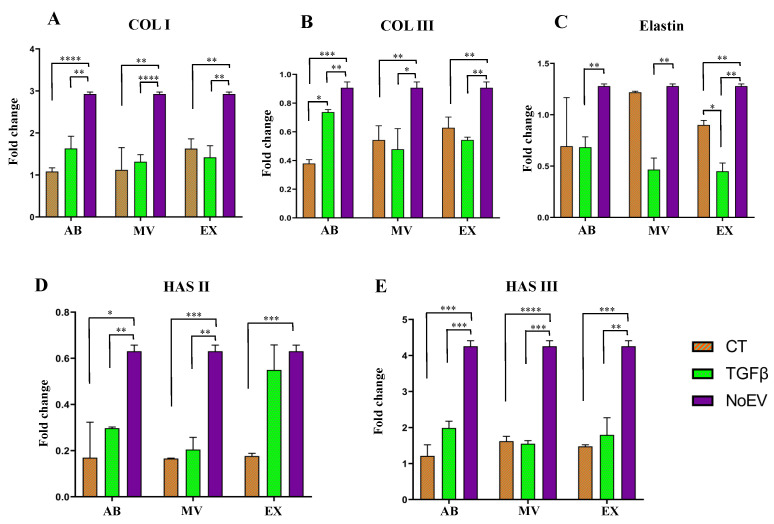
Induction of ECM gene expression in fibroblasts by EVs from TGFβ-stimulated UCMSCs and normal UCMSCs. There was no difference in COL I, COL III, Elastin, HAS II, and HAS III mRNA levels in dermal skin fibroblasts between EVs from TGFβ-stimulated UCMSCs and normal UCMSCs. Only COL III and Elastin in fibroblasts stimulated by ABs from TGFβ-stimulated UCMSCs and EXs from normal UCMSCs expressed higher levels compared to the normal condition and TGFβ-stimulated condition, respectively. (**A**) mRNA levels of COL I. (**B**), mRNA levels of COL III, (**C**) mRNA levels of Elastin, (**D**) mRNA levels of HAS II, and (**E**) mRNA levels of HAS III detected in fibroblasts under stimulation of EVs. CT: control group (EVs from UCMSCs cultured in normal condition), TGFβ: EVs from TGFβ-stimulated UCMSC group, NoEV: EV-depleted cell culture media, AB: apoptotic body, MV: microvesicle, EX: exosome. N = 3, * indicates *p* < 0.05, ** indicates *p* < 0.01, *** indicates *p* < 0.001, **** indicates *p* < 0.0001.

**Figure 5 biomedicines-10-01810-f005:**
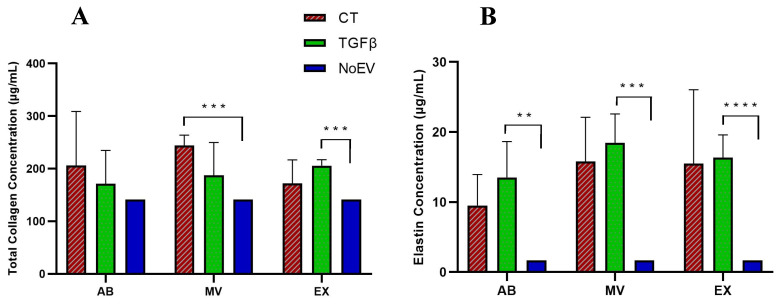
Collagen and elastin secretion by fibroblasts under stimulation of EVs from TGFβ-stimulated UCMSCs and normal UCMSCs. (**A**) Total collagen measured by ELISA assay indicated that MVs from the normal UCMSCs and EXs from TGFβ-stimulated UCMSCs stimulated collagen secretion by fibroblasts. (**B**) Elastin amount measured by ELISA assay indicated that all ABs, MVs, and EXs from TGFβ-stimulated UCMSCs stimulated fibroblasts to release elastin during 2D culture. CT: control group (UCMSCs cultured in normal condition), TGFβ: TGFβ-stimulated UCMSC group, NoEV: EV-depleted cell culture media, AB: apoptotic body, MV: microvesicle, EX: exosome. N = 3, ** indicates *p* < 0.01, *** indicates *p* < 0.001, **** indicates *p* < 0.0001.

**Table 1 biomedicines-10-01810-t001:** Primer sequences of COL I, COL III, HAS II, HAS III, Elastin, and GAPDH.

Gene name	Primer	Sequence	Size (bp)
COL I	Forward	CCTCAAGGGCTCCAACGAG	117
	Reverse	TCAATCACTGTCTTGCCCCA	
COL III	Forward	GAAGGGCAGGGAACAACTTG	243
	Reverse	TTTGGCATGGTTCTGGCTTC	
Elastin	Forward	GGCCATTCCTGGTGGAGTTCC	106
	Reverse	AACTGGCTTAAGAGGTTTGCCTCCA	
HAS II	Forward	ATGGGCAGAGACAAAT CAGC	249
	Reverse	GGCTGGGTCAAGCATAGTGT	
HAS III	Forward	AGCACCTTCTCGTGCATCAT	159
	Reverse	CTCCAGGACTCGAAGCATCT	
GAPDH	Forward	GGTGTGAACCATGAGAAGTATGA	123
	Reverse	GAGTCCTTCCACGATACC AAAG	

## Data Availability

Data are contained within the article.

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
