# Peer review of "Effects of Extracellular Vesicles Secreted by TGFβ-Stimulated Umbilical Cord Mesenchymal Stem Cells on Skin Fibroblasts by Promoting Fibroblast Migration and ECM Protein Production"

_biomedicines, 2022, doi:10.3390/biomedicines10081810_

Round 1
Reviewer 1 Report
The article by Duc Minh Vu and co-authors is devoted to the study of the effect of preconditioning of mesenchymal stromal cells from umbilical cord blood with transforming growth factor beta on the subpopulation composition of microvesicles and their effect on the functional potential of dermal fibroblasts, including aging. According to the data obtained from the conditioned medium of mesenchymal stromal cells of umbilical cord blood pretreated with TGF beta, as expected, the authors obtained 3 main subpopulations of microvesicles, which differed in size, expression level of differentiation clusters and morphology, which is not something new. The strong side of the work should include: 1) study of growth factors involved in skin repair/regeneration; 2) assessment of the effect of different subpopulations of microvesicles on the functional activity of fibroblasts. The weak side of the article should include: 1) the lack of comparison of the effect of the conditioned medium itself from the mesenchymal stromal cells of umbilical cord blood pretreated with TGF beta. There is evidence of superiority in the therapeutic potential of conditioned media from microvesicles. 2) lack of explanation of differences in the expression of extracellular matrix components and the level of production.
Author Response
We thank the reviewer for your contribution to improving the manuscripts.
Regarding the weak side of the manuscript mentioned by the reviewer, we have discussed it further in the discussion section in order to provide the reader with these points below:
“Additionally, the conditioned medium from MSC cultures has been reported for their therapeutic roles in cell proliferation and migration and ECM protein production [8] but not reported in this current study. Therefore, it should be more comparisons in roles of EVs and conditioned mediums to understand components affecting the bioactivities of the treated cells”. Page 11, line 27-31.
“The inverse expression of mRNA and protein levels of ECM components may reflect a hidden mechanism that promotes ECM protein production, particularly collagen, for example, microRNAs as their contribution to the regulation of mRNA translation. However, this hypothesis requires more careful investigation”. Page 11, line 49-53.

Reviewer 2 Report
The manuscript presents of three subpopulations of extracellular vesicles on fibroblasts in vitro. The data presented indicate the potential roles of EVs from UCMSCs in protecting skin from aging by promoting protein production of extracellular membrane proteins.
Remarks:
The title should be modified; “Skin Rejuvenation” suggests an in vivo study or at least study on a model skin while the study concerns only fibroblasts in vitro.
Fig. 1B Please label AB, MV and EX
Fig. 2 captions: please use plural rather than singular (apoptotic bodies etc.)
A linguistic check would be advisable; some not optimal terms are sometimes used.
Lines 47-49: “Factors causing cutaneous aging to induce collagenase production by human skin fibroblasts, which reduces collagen production and exposes the skin to wrinkles”, please correct this sentence
Lines 65-68: “Regarding skin tissue, the beneficial 65 effects of stem cell derived EVs on reducing skin aging characteristics due to they pro-66 moted migration and proliferation of epidermal cells and dermal fibroblasts, formation 67 and repair of blood vesicles”, please correct the sentence; the sentences lack the verb.
Author Response
RESPONSE TO REVIEWER 1
- Comments:
The manuscript presents of three subpopulations of extracellular vesicles on fibroblasts in vitro. The data presented indicate the potential roles of EVs from UCMSCs in protecting skin from aging by promoting protein production of extracellular membrane proteins.
Remarks:
The title should be modified; “Skin Rejuvenation” suggests an in vivo study or at least study on a model skin while the study concerns only fibroblasts in vitro.
Response:
Thank you the reviewer for this suggestion. We have revised the manuscript title from “Effects of Extracellular Vesicles Secreted by TGFβ-Stimulated Umbilical Cord Mesenchymal Stem Cells on Skin Rejuvenation through Promoting fibroblast Migration and ECM Protein Production” to “Effects of Extracellular Vesicles Secreted by TGFβ-Stimulated Umbilical Cord Mesenchymal Stem Cells on Skin Fibroblasts through Promoting Cell Migration and ECM Protein Production”.
- Comment:
Fig. 1B Please label AB, MV and EX
Fig. 2 captions: please use plural rather than singular (apoptotic bodies etc.)
Response:
We have corrected these label AB, MV and EX and all plural errors in Figure 1 Figure 2.
- Comment:
A linguistic check would be advisable; some not optimal terms are sometimes used.
Lines 47-49: “Factors causing cutaneous aging to induce collagenase production by human skin fibroblasts, which reduces collagen production and exposes the skin to wrinkles”, please correct this sentence
Lines 65-68: “Regarding skin tissue, the beneficial 65 effects of stem cell derived EVs on reducing skin aging characteristics due to they pro-66 moted migration and proliferation of epidermal cells and dermal fibroblasts, formation 67 and repair of blood vesicles”, please correct the sentence; the sentences lack the verb.
Response:
We would like to thank the reviewer for this. We have corrected the sentences as below:
“Factors causing cutaneous aging induce collagenase production by human skin fibroblasts, which reduces collagen production and exposes the skin to wrinkles”,
And:
“Regarding skin tissue, the beneficial effects of stem cell-derived EVs on reducing skin aging characteristics are that they could promote migration and proliferation of epidermal cells and dermal fibroblasts and the formation and repair of blood vesicles”.
And we have scanned through the manuscript for typos and grammar errors.

This manuscript is a resubmission of an earlier submission. The following is a list of the peer review reports and author responses from that submission.